# Waning of SARS-CoV-2 Vaccine Effectiveness in COPD Patients: Lessons from the Delta Variant

**DOI:** 10.3390/vaccines11121786

**Published:** 2023-11-29

**Authors:** Lörinc Polivka, Istvan Valyi-Nagy, Zoltan Szekanecz, Krisztina Bogos, Hajnalka Vago, Anita Kamondi, Ferenc Fekete, Janos Szlavik, György Surjan, Orsolya Surjan, Peter Nagy, Zsuzsa Schaff, Zoltan Kiss, Cecilia Müller, Miklos Kasler, Veronika Müller

**Affiliations:** 1Department of Pulmonology, Semmelweis University, 1085 Budapest, Hungary; polivka.lorinc@semmelweis.hu; 2South-Pest Hospital Centre, National Institute for Infectiology and Hematology, 1097 Budapest, Hungaryszlavikjanos@dpckorhaz.hu (J.S.); 3Department of Rheumatology, University of Debrecen, 4032 Debrecen, Hungary; szekanecz.zoltan@med.unideb.hu; 4National Korányi Institute of Pulmonology, 1122 Budapest, Hungary; bogos@koranyi.hu; 5Heart and Vascular Centre, Semmelweis University, 1122 Budapest, Hungary; vago.hajnalka@kardio.sote.hu; 6National Institute of Mental Health, Neurology and Neurosurgery, 1145 Budapest, Hungary; kamondianita@gmail.com; 7Heim Pál National Pediatric Institute, 1089 Budapest, Hungary; efekete@heimpalkorhaz.hu; 8National Public Health Center, 1097 Budapest, Hungary; surjan.gyorgy@nnk.gov.hu (G.S.); surjan.orsolya@nnk.gov.hu (O.S.); muller.cecilia@nnk.gov.hu (C.M.); 9National Institute of Oncology, 1122 Budapest, Hungary; peter.nagy@oncol.hu; 10Department of Pathology and Forensic Medicine, Semmelweis University, 1091 Budapest, Hungary; schaff.zsuzsa@med.semmelweis-univ.hu; 112nd Department of Internal Medicine and Nephrological Center, University of Pécs, 7624 Pécs, Hungary; dr.zoltan.kiss.privat@gmail.com; 12Ministry of Interior, 1051 Budapest, Hungary; miniszter@emmi.gov.hu

**Keywords:** COVID-19, COPD, vaccine effectiveness, VE, waning

## Abstract

Although the COVID-19 pandemic is profoundly changing, data on the effect of vaccination and duration of protection against infection and severe disease can still be advantageous, especially for patients with COPD, who are more vulnerable to respiratory infections. The Hungarian COVID-19 registry was retrospectively investigated for risk of infection and hospitalization by time since the last vaccination, and vaccine effectiveness (VE) was calculated in adults with COPD diagnosis and an exact-matched control group during the Delta variant of concern (VOC) wave in Hungary (September–December 2021). For the matching, sex, age, major co-morbidities, vaccination status, and prior infection data were obtained on 23 August 2021. The study population included 373,962 cases divided into COPD patients (age: 66.67 ± 12.66) and a 1:1 matched group (age: 66.73 ± 12.67). In both groups, the female/male ratio was 52.2:47.7, respectively. Among the unvaccinated, there was no difference between groups in risk for infection or hospitalization. Regarding vaccinated cases, in the COPD group, a slightly faster decline in effectiveness was noted for hospitalization prevention, although in both groups, the vaccine lost its significant effect between 215 and 240 days after the last dose of vaccination. Based on a time-stratified multivariate Cox analysis of the vaccinated cases, the hazard was constantly higher in the COPD group, with an HR of 1.09 (95%: 1.05–1.14) for infection and 1.87 (95% CI: 1.59–2.19) for hospitalization. In our study, COPD patients displayed lower vaccine effectiveness against SARS-CoV-2 infection and hospitalization but a similar waning trajectory, as vaccines lost their preventive effect after 215 days. These data emphasize revaccination measures in the COPD patient population.

## 1. Introduction

When vaccination is available, it is considered one of the best methods against viral infections, as, in most cases, it provides immunity to prevent or reduce the severity of the illness caused by the virus [1]. In addition, widespread vaccination can lead to herd immunity, where a high percentage of a population is immune to the virus, making viral spreading more difficult [1]. Overall, vaccination is a safe, effective, and important tool in controlling and reducing the impact of a great number of viral infections, including SARS-CoV-2-associated coronavirus disease-2019 (COVID-19). There are currently several different types of COVID-19 vaccines available, including mRNA vaccines (e.g., Pfizer-BioNTech and Moderna), vector vaccines (such as AstraZeneca), and protein subunit vaccines (such as Johnson & Johnson). The efficacy of the vaccines varies [2], but all have shown effectiveness in preventing severe illness, hospitalization, and death from COVID-19 [3]. Vaccination was a key tool in ending the global health emergency caused by the pandemic [4,5], while continuous surveillance is crucial as it helps to monitor the safety and effectiveness of vaccines after they have been approved and distributed to the public. Vaccine surveillance involves a range of activities, including collecting and analyzing data on adverse events, monitoring vaccine effectiveness (VE), and conducting ongoing risk–benefit assessments. Due to the limited time to produce and test the COVID-19 vaccines, the reassessment of reported VE and new calculations based on real-world data can be highly beneficial and complementary to previous results. Major distinctions of real-world studies compared to clinical trials are larger, achievable sample sizes, longer follow-up duration, no need for inclusion criteria, and the possibility to investigate different variants.

Chronic obstructive pulmonary disease (COPD) patients have an increased risk of airway infections, which comprise several viral causes including SARS-CoV-2 [6]. Both the Centers for Disease Control and Prevention (CDC) and the Global Initiative for Chronic Obstructive Lung Disease (GOLD) guidelines recommend annual revaccination for COPD patients with flu vaccines and against Pneumococci [7,8,9]. These guidelines also recommend considering additional vaccines, such as those for *Haemophilus influenzae* type b and measles, mumps, and rubella (MMR), based on individual patient risk factors. COVID-19 has had a devastating impact on this patient population, with higher morbidity and mortality rates reported as compared to non-COPD individuals [6,10]. As a response, in 2022, vaccination against COVID-19 was added to the GOLD recommendations for stable COPD [11]. Studies reporting VE and its change over time might further improve recommendations for COVID-19 vaccination in COPD patients and assist healthcare providers in making informed decisions about vaccination for these patients. To our knowledge, this is the first study reporting real-world VE in this patient group. In September 2023, Simon et al. stated in The European Respiratory Review, a well-regarded respiratory journal that is affiliated with the European Respiratory Society (ERS), that “there are no studies investigating COVID-19 vaccine effectiveness explicitly in COPD patients”, further strengthening the novelty of our results [12].

Our aim was to assess VE waning over time and the protective effect against infection or COVID-19 hospitalization of booster vaccinations in COPD patients using retrospective data. We aimed to provide evidence for the suggestion of an ideal time for revaccination for this specific patient population.

## 2. Materials and Methods

### 2.1. Study Population and Definitions

The study population included Hungarian residents aged 18 to 100 years, with a co-morbidity of COPD and a matched control group of non-COPD cases. COPD was defined as two occurrences of corresponding ICD-10 codes (J40-44, J47) in outpatient or inpatient claims data since 1 January 2013, using the second occurrence as the date of diagnosis. For the matching, we used one-to-one exact matching for sex, age (±1 year), co-morbidities (heart failure, history of stroke, history of ischemic heart disease, type 2 diabetes, immunosuppression, chronic kidney disease, and malignancy), vaccination status, and prior infection. ICD-10 codes of co-morbidities are available in Appendix A. The data were collected from the Hungarian-COVID-19 Registry made by the National Public Health Center (NPHC) and the National Health Insurance Fund Manager (NHIFM). This database and the case definitions are the same as published in an earlier analysis by this workgroup [13,14], containing anonymized data of all Hungarian citizens with a valid national insurance number (TAJ) of the date and type of COVID-19 vaccinations, the date and result of COVID-19 tests, the date of hospitalization, and general health information (e.g., known co-morbidities).

Individuals were classified as primary vaccinated if at least 14 days had passed since the administration of the second dose of BNT162b2 (Pfizer-BioNTech, Mainz, Germany), HB02 (Sinopharm, Beijing Bio-Institute of Biological Products Co Ltd, Bejing, China), Gam-COVID-Vac (Sputnik-V, Gamaleja Institute, Moscow, Russia), AZD1222 (AstraZeneca, Cambridge, United Kingdom), or mRNA-1273 (Moderna, Cambridge, MA, USA) vaccines, or only one dose of Ad26.COV2.S (Janssen, Leiden, Netherlands). Individuals were classified as boost vaccinated if at least 14 days had passed after the administration of any of the above-mentioned vaccines while already being primary vaccinated. At any time point during the study period, the unvaccinated control population included individuals who had not received any dose of any COVID-19 vaccine type beforehand. During the period between 23 August 2021 and 5 January 2022, the incidence of infection and vaccination status were recorded, and due to the non-availability of data after 2022, during the period between 13 September 2021 and 31 December 2021, COVID-19-related hospitalizations were recorded. Based on these data, the two investigated outcomes were defined as infection (reported in the National Public Health Center) and COVID-19 hospitalization, which was considered in cases with a positive PCR or antigen test at a maximum of 21 days before or 5 days after hospitalization.

### 2.2. Statistical Analysis

To count incidence rates for both outcomes, we used the ratio of affected cases and the population at risk at given periods after the last vaccination. VE was calculated as 1 min the rate of incidence rates of outcome in investigated populations and the unvaccinated population throughout the investigated period. For all VE values, exact 95% confidence intervals were calculated (reported using square brackets in tables). Cases with known previous infection were excluded when calculating risk rates and VE for infection or COVID-19 hospitalization. For the calculations of risks and VE, Python-based queries were used, with the help of the SciPy library [15]. For the calculation of hazard ratios (HR) for both of the above-mentioned outcomes, we used a time-stratified Cox-proportional hazard model of two factors (COPD vs. control and RNA vs. other vaccination type) and their interaction. For the calculations involving Cox-regression models, SPSS software version 27.0.1.0. (IBM, Armonk, NY, USA) was used.

The study was approved by the Central Ethical Committee of Hungary (OGYÉI/10296-1/2022 and IV/1722-1/2022/EKU).

## 3. Results

### 3.1. Patient and Population Characteristics

According to the given definition, 189,998 COPD cases fulfilled the criteria, of which 186,981 cases could be matched with non-COPD control cases with the exact matching method. The characteristics of the two groups at the start of the observation period (23 August 2021) are shown in Table 1. In Table 2, the outcomes (cumulative incidence) of the two groups by vaccination status with corresponding *p* values are visible.

### 3.2. Assessment of Vaccine Effectiveness throughout the Delta VOC Wave

The average VE during the whole observational period was only significantly different between the two groups for hospitalization in primary vaccinated cases, excluding those with known prior infection. The VE by vaccination status and groups is presented in Table 3, with the average days since the last vaccination until censoring (infection or end of observation). To investigate the VE over time, first daily risks for both outcomes in the unvaccinated independent of previous vaccinations risks by days since the last received vaccine dose were computed, and based on this VE by 15-day and then 5-day intervals were calculated. The daily risks for the unvaccinated are shown in Table 4, and those for the vaccinated are plotted in Figure A1 and Figure A2 and can be found in the Appendix A.

The calculated VE against both outcomes at 15-day intervals is shown in Figure 1, and VE against COVID-19 hospitalization at 5-day intervals is shown in Figure 2.

In Figure 3, only the suspected linearly declining period (between 145 and 220 days after the last vaccine dose received) is shown with fitted linear trendlines. Based on the trendlines in the control group, VE against COVID-19 hospitalization decreases by 3.5 percentage points, while in the COPD group, it decreases by 4.5 percentage points every 5 days. However, this is clinically non-significant as, after 215 days in both groups, the 95% CI for VE included 0%, meaning there is no clear evidence of benefit compared to the unvaccinated. As throughout these calculations the control group always seemed to have better VE, a time-stratified multivariate regression model was applied for all patients, including not only the group variable but, based on our previous results, a vaccine-type variable (RNA-based vs. other) as well [2]. The calculated HRs are shown in Table 5.

## 4. Discussion

### 4.1. Interpretation

Our data strongly suggest that COPD is associated with lower VE against both SARS-CoV-2 infection and hospitalization. While looking at the wave as a whole, no significant difference between the accounts of the two groups of unvaccinated and boost vaccinated groups was observed for COVID-19 hospitalization, while primary vaccinated COPD patients without booster showed significantly lower VE. With further investigation into VE dependent on time since the last vaccination in certain time windows (between 165 and 195 days after the last vaccination, when calculated by 15-day intervals, and between 185 and 195 days after the last vaccination, when calculated by 5-day intervals), there was a significant difference in VE against COVID-19 hospitalization between groups. With the time-stratified multivariate model, this difference was further confirmed and generalized for the whole period, but a clinically significant difference in waning was not observed between groups.

COPD patients were always considered to be at higher risk for severe outcomes of COVID-19 infection [15,16] and were a group that needed special attention during the pandemic [6]. Our result of impaired VE highlights their vulnerability. It is important to add, however, that this difference could be an effect of another COPD-associated patient characteristic, which could not be investigated in a populational study but caused differences in VE, like a smoking habit, which is known to be different between COPD patients and the general population and was shown to have an effect on immunization by vaccines [17]. Booster vaccinations or revaccinations later might be beneficial for this group, as was similarly shown in one of our previous studies for the elderly [14]. Additionally, it is important to note that, in this study, we included COPD patients of all severity. Differences between two COPD patients in different stages could be more substantial than between an early, mild COPD patient and a non-COPD patient. Because of the progressive nature of the disease, it would be beneficial to investigate VE in relation to COPD severity in later studies.

This study aims to help answer the question regarding the duration of protection provided by COVID-19 vaccines [18,19,20]. We used a similar approach to report robust retrospective datasets to the study by Nordström et al., which investigated the total Swedish population and showed progressively waning VE against infection of any severity across all subgroups [21]. Compared to their results, we observed a longer period of protection against severe disease and estimated the same start of decline in VE at around 4 months. Our study provides additional insight that there is no significant difference in VE decline between the COPD and non-COPD populations. This is partly a conformation and extension of the observation by Southworth et al. of similar immune responses in COPD patients occurring, on average, 100 days after vaccination compared to healthy subjects, and translates to real-world outcomes as well [22].

### 4.2. Limitations and Special Approach

Due to the real-life data-based retrospective nature of our study, it is important to list limitations, such as the unknown confounding factors, data quality, and the fact that a report focusing only on the Hungarian population may not be generalizable to other populations or settings. We want to emphasize the role of those confounding factors in VE, which might be more frequent in the COPD group but were not matched. These limitations are strongly connected to the main strength of the study: that it is a nationwide (large sample size), comprehensive, register-based study. Another limitation is that we did not conduct a one-year follow-up, which gives a broader time window for COVID-19 infection confirmation while analyzing hospitalizations, as recommended by the European Centre for Disease Prevention and Control (ECDC) protocol for COVID-19 VE [23]. The use of the exact number of days since the last administered vaccine dose instead of the vaccine status as a time-dependent variable, as recommended in the ECDC protocol, was to compensate for the ongoing booster vaccination campaign during the observation period and is thought to provide special insight into the change in VE over time.

We want to emphasize that our calculations excluded individuals who had a verified previous infection of SARS-CoV-2. While our study focuses on the initial vaccine effectiveness, we acknowledge the need for future research to assess vaccine effectiveness in individuals who have previously been infected, especially as reinfection with the virus has been reported even before the approval of the first vaccines [24].

### 4.3. Use for the Results of the Study after the End of the Pandemic

The waning of VE may be an important measure while planning revaccinations in the future based on previous practices concerning other vaccine-preventable viral diseases (e.g., seasonal flu) [25]. The results of this study could be a reference for later data, as VE can be at its lowest value during an outbreak [26]. The fact that this study showed around a 4-month period of protection with a following 3-month period of decline in effectiveness until there was no significant difference compared to the unvaccinated in a population that has access to several different types of vaccines could be a good estimate on when to start revaccinations, even in a population at increased risk of severe disease, such as patients with COPD. Additionally, the results of the time-stratified hazard models, showing a significantly higher risk for infection and hospitalization, can be used by public health efforts to prioritize COVID-19 revaccinations for this frailer patient group.

Finally, we want to emphasize the need for further research on vaccine effectiveness in COPD patients. All around the world, and in Hungary also, as of 2022, the Omicron VOC has profoundly changed the pandemic. Less pulmonary complications were observed, and patients and stakeholders introduced less strict preventative measures for viral spread. In light of these evolving circumstances and the use of new vaccines targeting other SARS-CoV-2 variants [27], there arises a compelling need for new studies to further explore and validate the findings of this paper.

## Figures and Tables

**Figure 1 vaccines-11-01786-f001:**
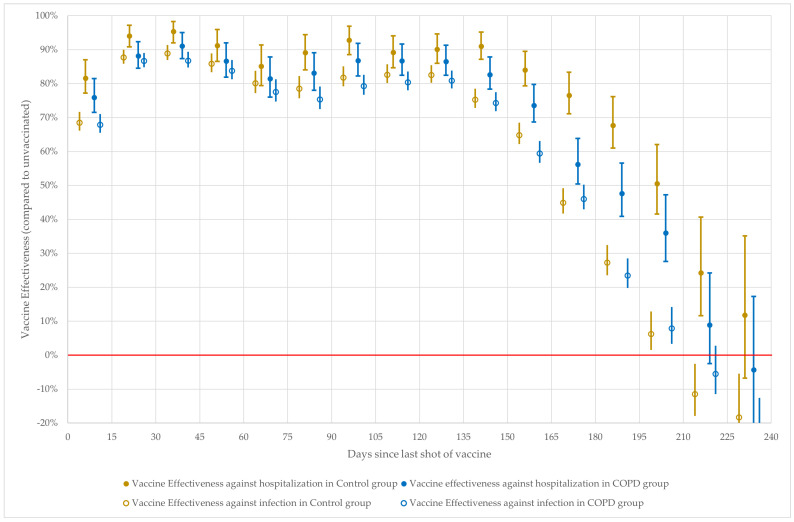
VE against infection and COVID-19 hospitalization by 15-day time intervals since last given vaccination dose.

**Figure 2 vaccines-11-01786-f002:**
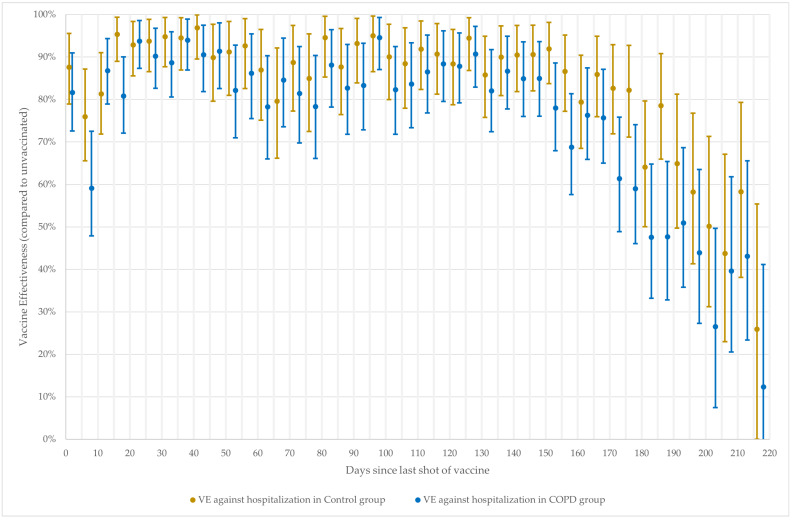
VE against COVID-19 hospitalization by 5-day time intervals since last given vaccination dose.

**Figure 3 vaccines-11-01786-f003:**
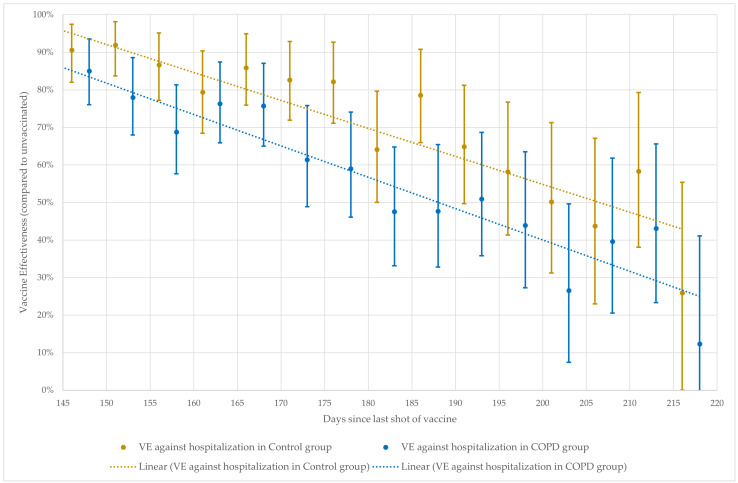
VE against COVID-19 hospitalization by 5-day time intervals between 145 and 220 days after the last vaccination dose.

**Table 1 vaccines-11-01786-t001:** Group characteristics at the start of the observation period.

	Matched Control	COPD
Sex		
Female	97,530 (52.2%)	97,530 (52.2%)
Male	89,451 (47.8%)	89,451 (47.8%)
Age (years) average ± SD	66.73 ± 12.67 *	66.67 ± 12.66 *
Comorbidities		
Heart failure	18,485 (9.9%)	18,485 (9.9%)
Acute myocardial infarct	2219 (1.2%)	2219 (1.2%)
Peripheral vascular disease	22,108 (11.8%)	22,108 (11.8%)
History of angina pectoris	27,545 (14.7%)	27,545 (14.7%)
History of stroke	15,426 (8.3%)	15,426 (8.3%)
Type 2 diabetes mellitus	43,872 (23.5%)	43,872 (23.5%)
Type 1 diabetes mellitus	126 (0.1%)	126 (0.1%)
Any malignancy	28,249 (15.1%)	28,249 (15.1%)
Immunosuppressed state	12,198 (6.5%)	12,198 (6.5%)
Chronic kidney disease	10,356 (5.5%)	10,356 (5.5%)
Immune status		
Unvaccinated	38,017 (20.3%)	38,017 (20.3%)
Primary vaccination	119,676 (64%)	119,676 (64%)
Boost vaccination	15,705 (8.4%)	15,705 (8.4%)
Prior infection	13,583 (7.3%)	13,583 (7.3%)

* The difference between the average age is not statistically significant, with a *p* value of 0.172 for two-sided homoscedastic students’ *t*-test.

**Table 2 vaccines-11-01786-t002:** Outcomes and vaccination status of infected and hospitalized patients during the study period.

	Matched Control	COPD	*p*
	n = 186,981	n = 186,981	
Infected	7398 (4.0%)	8025 (4.3%)	<0.001
Unvaccinated	2991 (40.4%)	3095 (38.6%)	
Primary vaccinated	3580 (48.4%)	3921 (48.9%)	0.031
Boost vaccinated	684 (9.2%)	817 (10.2%)	
Had prior infection	143 (1.9%)	192 (2.4%)	0.050
Hospitalized (COVID-19 hospitalization)	1278 (0.7%)	1842 (1.0%)	<0.001
Unvaccinated	726 (56.8%)	844 (45.8%)	
Primary vaccinated	456 (35.7%)	825 (44.8%)	<0.001
Boost vaccinated	88 (6.9%)	152 (8.3%)	
Had prior infection	8 (0.6%)	21 (1.1%)	0.141

The *p* values shown are calculated by comparing the two groups with Pearson Chi-Square statistic.

**Table 3 vaccines-11-01786-t003:** Vaccinee effectiveness by vaccination status and groups.

	Primary Vaccinated		Boost Vaccinated	
	**Matched Control**	**COPD**	**Matched Control**	**COPD**
VE against Infection	46.78%	45.58%	86.11%	83.6%
	[43.48–49.94%]	[42.30–48.74%]	[84.65–87.50%]	[82.02–85.10%]
VE against COVID-19	72.07%	58.01%	92.64%	88.81%
Hospitalization	[67.51–76.22%]	[52.59–63.01%]	[90.45–94.54%]	[86.30–91.06%]
Average No. days since last vaccination at censoring time	240.67 ± 57.82	239.06 ± 59.79	69.3 ± 45.16	69.89 ± 45.74

**Table 4 vaccines-11-01786-t004:** Daily risks amongst unvaccinated subgroups for infection and hospitalization.

	Matched Control	COPD
Daily risk for infection	64.71 [61.36–67.16]/100,000	67.73 [64.25–70.28]/100,000
Daily risk for COVID-19 hospitalization	15.84 [14.2–17.07]/100,000	18.66 [16.86–20.02]/100,000

**Table 5 vaccines-11-01786-t005:** Hazard ratios with 95% CI and significance level in a time-stratified Cox-regression model for infection and COVID-19 hospitalization, respectively.

	Infection		COVID-19 Hospitalization	
	**HR**	* **p** *	**HR**	* **p** *
mRNA vaccine	0.66 [0.619–0.703]	<0.001	0.809 [0.68–0.962]	0.017
Group COPD	1.073 [1.008–1.142]	0.028	1.867 [1.592–2.190]	<0.001
Interaction	1.034 [0.953–1.123]	0.420	0.889 [0.724–1.093]	0.266

## Data Availability

The original contributions presented in the study are included in the article. Further inquiries can be directed to the corresponding author.

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
