# Peer review of "Waning of SARS-CoV-2 Vaccine Effectiveness in COPD Patients: Lessons from the Delta Variant"

_vaccines, 2023, doi:10.3390/vaccines11121786_

Round 1
Reviewer 1 Report
Comments and Suggestions for Authors
The authors organized a manuscript entitled “Waning of SARS-CoV-2 Vaccine Effectiveness in COPD Patients: Lessons from the Delta Variant”. Overall, this manuscript signifies an effort to provide evidence regarding the clinical efficacy of SARS-CoV-2 vaccines in the Hungarian population and discuss the duration of protection of vaccines against SARS-CoV-2, based on the reality in the field. I found that the topic of current MS is adequate and acceptable for the journal’s scope. Even more, the manuscript addresses an important topic within the COVID-19 pandemic situation. The manuscript is well-prepared and it is worthy of publication. However, to this end, I have some concerns/comments related to the contents written in the manuscript.
1. While the authors have argued for the importance of this topic, the novelty of this manuscript remains overlooked. The introduction section written in the manuscript is quite short (and lacking of persuasion) to provide insights into why this manuscript is important.
2. Authors' results indicate that COPD is associated with lower vaccine effectiveness against both SARS-CoV-2 infection and hospitalization. With many variants of SARS-CoV-2 and vaccine platforms that have been described, the authors need to discuss whether this is based on the vaccine platforms or whether this is based on the presence of special groups of patients such as the immunodeficiency patients and pregnant women, in the tested groups. For example, Kumar et al. 2021 (http://dx.doi.org/10.1016/j.jiph.2021.04.005) have argued the pregnant women are considered a highly vulnerable group. These patients may have limited or impaired ability to mount adaptive immune responses, especially the immunodeficient ones, due to impaired activation of adaptive immune responses. Would this population be able to mount proper immune responses after being vaccinated? Or perhaps this population experiences a waning of immune protection faster than other groups?
3. The authors suggested that revaccination is important. While I agree with this suggestion, the authors may need to strengthen their discussion about the potency of vaccines in the mitigation of SARS-CoV-2 reinfection. Please kindly include the reference Wang et al. 2021 (http://dx.doi.org/10.1136/jim-2021-001853) and Nainu et al. 2020 (https://doi.org/10.1080/21645515.2020.1830683) in the discussion of such matter.
4. The authors observe that while primary vaccinated COPD patients without boosters showed significantly lower VE, there was no significant difference between the two groups in outcomes of unvaccinated and boosted vaccinated groups observed for COVID-19 hospitalization. Why did this happen?
5. With limited time to produce and test the COVID-19 vaccine, the authors need to elaborate their discussion on what factors determine vaccine effectiveness. What are the criteria that need to be considered in the safety and effectiveness assessment of vaccine candidates? How are these criteria assessed and how long shall they be assessed prior to administration to the targeted population? Surely, this will improve the discussion section to support the authors' suggestion for revaccination.
6. All figures shall be re-prepared to enhance their quality. Labels written in all figures are not visible enough. The authors may need to think about whether changing the color of choice would improve the visibility of figures.
7. Minor: Ref 25 (line 178 page 7) is not written properly. Please check thoroughly in the manuscript for similar mistakes.
Finally, I recommend that the authors shall address all comments/concerns above prior to consideration of publication in this journal.
Comments on the Quality of English LanguageThe manuscript is well-written and well-prepared.
Author Response
See our response attached!

Reviewer 2 Report
Comments and Suggestions for Authors
By comparing vaccine effectiveness (VE) in COPD patients and age and sex matched controls, this large study sought to provide evidence for the choice of the ideal time between revaccinations for COPD patients. Data were collected from the Hungarian-COVID-19 Registry compiled by the National Public Health Center and the National Health Insurance Fund Manager. Commendable effort and planning resulted in carefully matched age, sex and comorbidity data enabling this interesting study.
An important objective for the study was to provide improved vaccination guidelines for COPD patients by comparing VE in COPD patients and comparing their responses with data for non-COPD individuals.
The introduction is brief, but it effectively provides an adequate background for appreciating the major objective of the study.
The study population and definitions are outlined clearly in 2.1 of Materials and Methods.
Statistical analysis appears to have been carried out using appropriate and well-chosen tests that with the large dataset, should enable statistically valid conclusions to be drawn.
Table 1 Summarises the group characteristics at the start of the observation period. The row headings Title 2 and Title 3 should be clarified because at first glance they are difficult to comprehend. Perhaps re-name the rows and add an additional explanatory note to the text below the Table.
In Table 2 the p values are clear but adding the test employed below the table would help the reader appreciate the analysis. Presume this is Homoscedastic Students T-test?
Fig 1 clearly enables an appreciation of the decline in VE for both control and COPD groups, both approaching zero at around day 210.
In Fig 3 the suspected linearly declining VE between day 145 and day 220 are graphically presented clearly, however the data are not clinically significant signifying no clear evidence of benefit compared to the unvaccinated.
Figure A2 should be incorporated into the body of the paper since that graph seems to clearly show the daily risk for the COVID hospitalization is biased toward the COPD patients.
This should be discussed in support of the text L138 to L143. Fig A2 seems to indicate that beyond about day 130 the COPD patients are over-represented in the daily risk for COVID hospitalization which seems to be in accord with other published studies. What I am suggesting is that the discussion L130 to L159 might profitably be restructured to stress the relationship between declining VE and increased risk for COVID hospitalization in COPD patients. I believe these interrelationships lend themselves to depiction in a flow diagram!
The limitations suggested by the authors are very useful and the suggested use of the time-stratified hazard models by public health authorities to prioritise COVID-19 vaccines to this COPD patient group is also very useful.
Impressive numbers of participant patients extracted from state medical records eg for COPD. Was severity of COPD diagnosis noted in this group? Did any diagnosis of COPD a trigger that all patients could be accepted?
A useful and interesting study that provides evidence for the waning of vaccine effectiveness and the implications for vulnerable groups, in this case patients diagnosed with COPD. It is a very useful study in that it draws upon records extracted from a large national health database.
L9 In both these groups, the female/male ratio was 52.2:47.7 respectively.
L10/11 Suggesting change to “Among the unvaccinated there was no difference between groups in risk …….” If this still conveys the meaning intended.
L16 “ ….COPD patients displayed a lower vaccine effectiveness….”
L21 - 22 “Vaccination is considered one of the best methods to protect against viral infections by providing, in most cases, immunity ….”
L30-31 “ …all have shown effectiveness in preventing ….”
L39- 41 CDC recommendations vaccinations for COPD patients
L47 delete “can”
L52 “We aimed to provide a suggestion…”. Better to state as “ We aimed to provide evidence for the ideal time….”
L85. “affected”
L93 “…a time-stratified Cox-proportional hazard model …
L64. “involving Cox-regression models, …”
In Table 1 explain clearly the meaning of Title 2 and Title 3.
Table 1 Clarify the difference between headings Title 2 and Title 3? I only see minor difference in mean age??? Maybe use a different term for the headings in lieu of Title 1 and Title 2. Maybe add a comment on how participants in the two groups were allocated!
Author Response
See our response attached!

Round 2
Reviewer 1 Report
Comments and Suggestions for Authors
The revised version is very much improved. All revised parts have been prepared in a proper manner.